# Comparison of Several Prediction Equations Using Skinfold Thickness for Estimating Percentage Body Fat vs. Body Fat Percentage Determined by BIA in 6–8-Year-Old South African Children: The BC–IT Study

**DOI:** 10.3390/ijerph192114531

**Published:** 2022-11-05

**Authors:** Lynn Moeng-Mahlangu, Makama A. Monyeki, John J. Reilly, Herculina S. Kruger

**Affiliations:** 1Physical Activity, Sport and Recreation Research Focus Area (PhASRec), Faculty of Health Sciences, North-West University, Potchefstroom 2531, South Africa; 2Physical Activity for Health Group, School of Psychological Sciences and Health, University of Strathclyde, Glasgow G1 1XQ, UK; 3Centre of Excellence for Nutrition, North-West University, Potchefstroom 2531, South Africa

**Keywords:** body composition, skinfold thickness, bioeletrical impedance analysis, adiposity, children

## Abstract

Body composition measurement is useful for assessing percentage body fat (%BF) and medical diagnosis, monitoring disease progression and response to treatment, and is essential in assessing nutritional status, especially in children. However, finding accurate and precise techniques remains a challenge. The study compares %BF determined by bioelectrical impedance analysis (BIA) and calculated from available prediction equations based on skinfolds in young South African children. A cross-sectional study performed on 202 children (83 boys and 119 girls) aged 6–8 years. Height and weight, triceps and subscapular skinfolds were determined according to standard procedures. %BF was determined with BIA and three relevant available equations. SPSS analyzed the data using paired samples tests, linear regression, and Bland–Altman plots. Significant paired mean differences were found for BIA and Slaughter (t_201_ = 33.896, *p* < 0.001), Wickramasinghe (t_201_ = 4.217, *p* < 0.001), and Dezenberg (t_201_ = 19.910, *p* < 0.001). For all of the equations, the standards for evaluating prediction errors (SEE) were above 5. The Bland–Altman plots show relatively large positive and negative deviations from the mean difference lines and trends of systematic under- and over-estimation of %BF across the %BF spectrum. All three equations demonstrated a smaller %BF than the %BF measured by BIA, but the difference was smallest with the Wickramasinghe equation. In comparison, a poor SEE was found in the three %BF predicted equations and %BF derived from BIA. As such, an age-specific %BF equation incorporating criterion methods of deuterium dilution techniques or ‘gold-standard’ methods is needed to refute these findings. However, in the absence of developed %BF equations or ‘gold-standard’ methods, the available prediction equations are still desirable.

## 1. Introduction

Increasing childhood obesity is a significant public health threat in the modern world [1,2]. Over the past 40 years, obesity in the age group 5–19 years old increased in all regions among girls and boys [1]. McPhee et al. [3] estimate that globally 381 million children are either overweight or obese. The upsurge is observed more in low- and middle-income countries [4,5], with urban areas being more affected [5], including in South Africa [6]. The relationship between excessive body fat and increased morbidity and mortality is well documented [7]. Early diagnosis of obesity is vital, including understanding the risk factors for obesity, given the fact that childhood obesity predicts obesity later in adulthood [2,7]. Conversely, preventative strategies require a better understanding of contributing factors and an accurate assessment of overweight and obesity [8]. 

The most common, easy-to-use, and cost-effective methods to assess overweight and obesity are based on anthropometric measures such as the body mass index (BMI): weight divided by height squared, and skinfolds. Scientific evidence on the misclassification of obesity using BMI exists, especially in children [1,8]. BMI does not distinguish between lean muscle from fat mass [9], nor does it accurately reflect visceral fat accumulation. Visceral fat is the likely culprit leading to most of the metabolic and clinical consequences of obesity [9]. Ideally, obesity should be determined based on the percent of body fat (%BF), which is considered a better marker for assessing excess adiposity or obesity [10,11,12].

Body composition measurement is useful for assessing %BF and medical diagnosis and monitoring disease progression and response to treatment [12]. It is further regarded as essential in assessing nutritional status, especially in children [6]. However, finding accurate and precise techniques remains a challenge [13]. The majority of the prediction equations used to assess body composition have been developed and cross-validated for Caucasians, notwithstanding comparative studies [14], and may be unsuitable for other ethnic groups.

Numerous methods are available for determining %BF in adults and children. However, some are costly, difficult to use, and time-consuming [12], including dual-energy X-ray absorptiometry (DEXA) and air-displacement plethysmography (ADP) [15]. The equipment used for such studies may be bulky, making them unsuitable for clinical settings and field research [16]. Additionally, it was found that DXA has a limitation in scanning obese individuals because they exceed the weight limitations or because their body size exceeds the scanning area [12]. In a study by Nicholson et al. [17] in 6-to14-year-old African American and White children, it was found that measurements of body fat calculated from the application of the Siri equation to ADP density measurements are equivalent to those obtained by DXA in boys. However, it significantly underestimated the body fat of girls by a small amount. Therefore, there is an ongoing need to investigate or compare other methods that can accurately determine %BF in a particular population groups. Bioelectrical impedance analysis (BIA) and skinfold-thickness as alternative methods that are rapid, noninvasive, simple, and less costly methods for evaluating body composition in field and clinical settings [18]. Both skinfold thickness [19] and BIA provide acceptable prediction %BF values in certain populations [18]. Furthermore, BIA is suitable for large epidemiological studies to assess body composition and predict %BF in healthy subjects [20]. 

BIA and skinfold thickness have been used for decades in different population groups, and their prediction equations have been validated against known reference methods, producing poor to good agreement results. Wan et al. [21] argued that, regardless of the uncertainty in the reference method, BIA might represent the ‘true’ average value for adiposity in Australian adolescents and suggested a need for more research work in the area. Nevertheless, no studies were found that compare prediction equations using skinfold thickness for estimating percentage body fat against BIA among South African children. The Slaughter et al. [22] equation uses the sum of triceps and subscapular skinfolds to predict %BF and has been validated in different populations, and inconsistent results were found [23]. Nonetheless, comparative studies for BIA and against skinfolds equations in South African children remain to be conducted. Therefore, an increasing number of scholars recommend the development of BIA and skinfold-thickness race-specific prediction equations for %BF and FFM to improve precision [12,23]. Hence, paucity still exists regarding comparative studies on predicted %BF for South African children. 

Therefore, this study focused on comparing %BF derived from BIA with body fatness derived from published skinfold-thickness equations (i.e., Slaughter et al. [22], Wickramasinghe et al. [24], and Dezenberg et al. [25]. Over the decades, various skinfold equations have been developed, with the majority of the prediction equations validated in older children aged 10–12 years old and in adults [22,26], using varying skinfold sites or prediction equations. For example, there have been studies on 11–13-year-old African-American, non-Black Hispanic, and non-Hispanic White (Caucasian) girls [17]; 65 to 103 year olds from Guatemala City [27], people aged 23.6–53.1 years from the Gateshead Millennium Study (GMS) [28]; Caucasian American adults aged 65 and older [29]; and 18–30-year-old Spanish people [30]. A comparative study on prediction equations was from India on 26-to-49-year-old people by Chahar [30], whereby a comparison for predicted percentage body fat (%BF) was derived from BIA with a Maltron BF 908 body composition analyzer, a skinfold equation given by Durnin and Womersley, and a body-mass-index-specific prediction formula given by Deurenberg, Weststrate, and Seidell. In this study, it was reported that BIA underestimated %BF when %BF was assessed by skinfold-thickness measurements and BMI methods. 

The skinfold prediction equations by Slaughter et al. [22] Wickramasighe et al. [24], and Dezenberg et al. [25] were selected because of the age range, the skinfold sites used in the validation studies, and the method used. Furthermore, it was noted that these three skinfold prediction equations have been validated and used in different populations and found to produce unreliable results with BIA [25,31]. However, González-Ruíz et al. [15] stated that the use of BIA for the estimation of fat-free mass (FFM) in healthy individuals in low-resources communities as an alternative should be used with caution, given the fact that BIA prediction models differ according to the sample characteristics in which they have been derived. Even though BIA is not a criterion method, comparative studies against its use for predicting %BF and existing skinfold predictions in South African children is hardly studied. This study, therefore, compares two secondary methods for %BF determined by BIA and calculated from available prediction equations based on skinfolds in South African children given the limited comparative studies in South African children. 

## 2. Materials and Methods

Study Design and Study Sample

A cross-sectional design was used, with a total of 202 children (boys = 83; 41.1% and girls = 119; 58.9%) aged 6 to 8 years old with a complete skinfold-thickness measurement from a sample of 299 children from a larger body composition using an isotope technique study (BC–IT study, Figure 1). The larger BC–IT study examined the relationship between objective (stable isotope, BIA) and indirect (anthropometric variables) measures of BC indices. The current study aims to compare the two field methods included in the BC–IT study. Therefore, more details about the study have been published elsewhere [32]. Briefly, the study was drawn from five primary schools randomly selected from a total of 26 primary schools in Tlokwe City in North-West Province, South Africa. The sample size was calculated using Open Epi software, Version 3 [33], in which Fleis’s [34] formulae for cross-sectional studies were applied to determine the appropriate sample size for a power of 0.80 and α-level of 0.05 at a CI of 95%. The power calculation was based on the primary hypothesis of a negative association between excessive fatness and physical activity and the odds of having excessive percentage body fat (%BF) in the inactive group, from which it was found that a minimum sample size of 297 was needed, based on an expected prevalence of combined overweight/obesity of 20% and physical inactivity of 30% in the children. Every third child on each class list was selected to participate in the study, but only those with signed parental informed consent forms who personally agreed to participate were finally included [32]. Subsequently, the Human Research Ethics Committee (HREC) of North-West University gave ethical approval to the study (ethic no: NWU-00025-17-A1). Parents of the children and the children gave consent and assent to participate in the study, respectively. Subsequently, children voluntarily participated in the study.

### 2.1. Measurements

#### Anthropometry

Weight was measured with a portable electronic scale (Beurer^®^ Ps07 Electronic Scale, Ulm, Germany) to the nearest 0.1 kg. Children were measured with light clothing and without shoes. Stature was measured by a Harpenden^®^ portable stadiometer with a perpendicular board to the nearest 0.1 cm. 

Triceps and subscapular skinfolds were taken with a Harpenden^®^ skinfold calliper [35] with a constant pressure of 10 g/mm^2^ to the nearest 0.2 mm on the left sides of the participant. All of the anthropometric measurements were done by a qualified Level I anthropometrist according to the standard procedures of International Society for the Advancement of Kinanthropometry (ISAK) procedures [35]. The body mass index for age z-score (BMI z) were computed using WHO AnthroPlus software and then were used for the classification of children as: normal-weight (BMI z −2 to +1 SDs; i.e., >5th percentile to <85th percentile equivalent to BMI of 18.5 through 24.9); thin or underweight (BMI Z =<−2 SDs; i.e., <5th percentile equivalent to BMI less than 18.5); overweight (BMI z (>+1 to 2 SD, i.e., 85th to <95th percentile equivalent to BMI of 25.0 through 29.9)); obese (>+2SD; i.e., ≥95th percentile equivalent to BMI of ≥30.0) for their age, according to the child growth reference for children 5–19 years old [36].

### 2.2. Body Fat Percentage by Bioelectrical Impedance Analysis (BIA)

Body composition was assessed using BIA (Bodystat1500MDD, MultiScan 5000, BodyStat). Participants were asked to remove socks, jewelery, and belts containing metal or metal-rimmed glasses. The electrodes of the BIA were placed on the dorsal surface of the right hand and foot at the metacarpals and metatarsals, respectively. Detection electrodes were placed at the pisiform prominence of the right wrist and the anterior surface of the ankle joint in accordance with standard procedures [37]. Participants were not allowed to eat or drink anything four hours prior to testing, to exercise within 12 h of the test, or to urinate within 30 min of the first test. 

Existing BIA Bodystat software with inbuild equations was used for the estimation of total body water (TBW). The Bodystat software produces an output specifying TBW in liters, FFM (fat-free mass) (kilograms), fat mass (FM) (kilograms), and %BF, as well as impedance, resistance, and reactance readings. The BIA prediction equations for TBW and FFM have been validated for use in Chinese, Lebanese, Malay, Filipino, and Thai children aged 8 to 10 years (948 participants) across a wide BMI range (12.2–34.9 kg/m^2^) [37,38]. 

Children were classified as underweight/underfat, normal, overfat, or obese based on the McCarthy %BF centile curves [16]. Standards for estimating prediction errors and interpretations are in accordance with Lohman as cited by Heyward and Stolarczyk [18]. The characteristics of skinfold-equation methods used for deriving %BF in children derived from Wickramasinghe, Slaughter, and Dezenberg are presented in Table 1.

#### Statistical Methods

Normality distribution of the data was determined using the Kolmogorov–Smirnov test and graphical methods, i.e., histograms and Q–Q plots. Descriptive statistics (mean and standard deviations (SD)) for normally distributed data were calculated. If data did not follow normal distribution, nonparametric techniques were used. For categorical variables, frequencies for percentages were calculated. We performed an independent t-test for normally distributed data to determine sex differences and a chi-squared test for age and BMI categories. Paired sample t-test, paired correlations, and differences were used to evaluate the relationship and difference between %BF obtained from BIA and the prediction equations. Furthermore, a paired t-test was used to determine the limits of agreement between %BF obtained from BIA and the selected prediction equations and regressions; Bland and Altman plots were computed. The limits of agreement (LOA) were defined as the mean ± 1.96 SD for the upper and lower limits. Analysis of variance (ANOVA) for multiple comparisons of Bonferroni post-hoc tests were performed to determine age-group differences. All statistical analyses were performed using the Statistical Package for the Social Science (SPSS). The level of significant differences was set at *p* < 0.05.

## 3. Results

Table 2 presents the descriptive characteristics of boys and girls. Significant sex differences were observed for %BF measured by BIA (*p* < 0.001), triceps skinfold (*p* < 0.001), and TBW (*p* = 0.001). No significant (*p* > 0.05) sex differences were found between age, weight, height, BMI, and BMI for age z-score. Based on BMI categories, out of 202, 15.3% (n = 31) of the children were overweight and 6.9% (n = 14) obese, with girls being more overweight (21.8%; n = 26) and obese (10.1%; n = 12) compared to the boys (overweight = 6.0%; n = 5) and obese (2.4%; n = 2). Underweight children were 14.9% (n = 30) for the total sample participants, with more boys being underweight (22.9%; n = 19) than the girls (9.2%; n = 11). It was evident that there are weight differences between boys and girls (22.73 ± 4.53 and 22.69 ± 4.15, *p* = 0.91, for boys and girls, respectively). Similarly, fat mass in boys and girls was significantly different using both the Slaughter et al. [22] and Wickramasinghe et al. [24] equations (*p* =< 0.001). Lastly, no significant sex differences (*p* = 0.35) in body fatness could be found with the Dezenberg et al. [25] prediction equation. 

Table 3 presents differences between %BF by BIA and several skinfold equations. Bioelectrical impedance analysis moderately and positively correlated with the Slaughter equation (r = 0.61, *p* < 0.001) and with the Wickramasinghe et al. [22] equation (r = 0.54, *p* < 0.001), and a low correlation was observed with the Dezenberg et al. [25] equation (r = 0.47, *p* < 0.001). There were significant mean differences between BIA and the Slaughter et al. [22] equation (t_201_ = 33.896, *p* < 0.001), the Wickramasinghe et al. [24] equation (t_201_ = 4.217, *p* < 0.001), and the Dezenberg et al. [25] equation (t_201_ = 19.910, *p* < 0.001). 

The differences and the limits of agreement between %BF obtained from BIA and various selected skinfold equations are described in Table 4 and Table 5, respectively. The means of %BF were 22.45 ± 5.1 and 26.94 ± 6.38 in boys and girls, respectively. 

All skinfold prediction equations showed a poor SEE above 5.0 based on the Lohnman standards for evaluating prediction-error classifications. However, %BF in boys compared with BIA with the Dezenberg et al. [22] equation showing a smaller mean ± SD difference of −5.13 ± 5.64 compared to the Slaughter equation, which showed mean ± SD of 11.92 ± 4.92. In girls, the Wickramasinghe et al. [24] equation showed the lowest mean ± SD, 0.52 ± 6.05, and the largest difference was with the Slaughter et al. [22] equation at −12.65 ± 5.35 (Table 5). 

The lowest limit of agreement was observed from the Slaughter et al. [22] equation in both boys and girls, and all prediction equations underpredicted %FM by different percentages ranging from 24% to 53% among boys and −1.9% to 47.4% among girls (Table 4). The Slaughter prediction equation showed the largest difference of 53% and 47.4%, among boys and girls, respectively. 

The mean difference between the calculated (Slaughter et al. [22] equation) and measured %BF were −12.65 ± 5.35 (Table 5). The difference between the two methods is large and indicates that the Slaughter equation most underestimates %BF measured by BIA (*p* < 0.0001). A slope of 0 would mean that the difference between the calculated %BF by the Slaughter et al. [22] equation minus measured %BF by BIA stays at the same level across the range of %BF from low to high (Figure 2a). Regression analysis demonstrated a negative slope (−0.329) that implies that there is a greater positive difference between calculated %BF by the Slaughter et al. [22] equation minus measured %BF by BIA at low %BF of the children (<20%), while the difference changes to a negative difference at high %BF. The differences are the smallest between 20–28 %BF, but there are many outliers.

The mean difference between the calculated (Dezenberg et al. [25] equation) and measured %BF were −7.28 ± 6.12. This is a smaller difference but still an underestimation of %BF compared to the %BF measured by BIA and significantly different from 0 (*p* < 0.0001). Figure 2b also shows a negative slope (standardized beta) of −0.203 (Figure 2b). The smaller slope indicates a better fit across the range of %BF from low to high. However, there is also a greater positive difference between the calculated %BF by the Dezenberg et al. [25] equation minus measured %BF by BIA at low %BF of the children (<18%). The difference changes to a negative difference at high %BF. It seems the differences are the smallest, between 18–25 %BF.

The mean difference between calculated (Wickramasinghe et al. [24] equation) and measured %BF = −1.92 ± 6.48 (Table 5). This is a small difference, but still, the difference is significantly different from 0 (*p* < 0.0001). Figure 2c shows the smallest slope compared to the other two figures, indicating a better fit across the range of %BF from low to high. Due to the small positive slope (0.108), there are almost as many negative as positive differences between calculated %BF by the Wickramasinghe et al. [24] equation minus measured %BF by BIA at low %BF of the children (<30%), while the difference changes to more positive differences at high %BF (>30%). The differences are relatively fairly stable across the spectrum from 12–40 %BF. Therefore, this equation showed the best agreement with %BF measured by BIA in the total group across the spectrum (smallest mean difference and smallest slope of the regression line). Only six observations are outside the limits of agreement.

There are relatively large positive and negative deviations from the mean difference lines and considerable variation in all three plots and trends of differences across the %BF spectrum, particularly when the Slaughter equation is applied. With all three equations, we found a smaller %BF than the %BF measured by BIA, but the difference is negligible with the Wickramasinghe et al. [24] equation in girls aged 5 to 15 years old. 

Comparatively, all the three predictive equations (Slaughter et al. [22], Wickramasinghe et al. [24], and Dezenberg et al. [25]) performed poorly on Bland–Altman plots. Additionally, the prediction equations varied in the slope of the regression line and the developed standards of evaluating prediction errors of body composition estimating %BF by Lohman.

## 4. Discussion

The study compared predicted %BF derived from two secondary methods of BIA and published equations (i.e., Slaughter et al., Wickramasinghe et al., and Dezenberg et al.). In comparison, a poor agreement (above Lohman SEE cut point of 5.0) between BIA measurements and the prediction equations by Slaughter et al. [22], Wickramasinghe et al. [24], and Dezenberg et al. [25] was observed. With all three equations, we found a smaller %BF measured by BIA, but the difference is smallest with the Wickramasinghe et al. [24] equation. 

When selecting a prediction equation to use, it is important to note the population from which the equations was derived from. Dezenberg et al. [25] indicated that the use of an equation developed for adults to predict %BF may result in overestimation in children. Furthermore, there are few available skinfold-prediction equations valid across populations. The skinfold-prediction equations selected for the current study were developed for use in children: Slaughter et al. [22], for 8-to-29-year-old African and Caucasian children and young adults; Dezenberg et al. [25], for 4-to-10.9-year-old Caucasian and African American children; and Wickramasighe et al. [24], for 5-to-15-year-old Asian children. The Slaughter equation has been used in different ethnic groups, producing high correlation coefficients (BMI, r = 0.78; *p* < 0.001) with BIA in a Mexican study (mean age 9.47 ± 1.55 years) [38]. 

The skinfold-prediction equations by Wickramasinghe et al. [24] and Slaughter et al. [22] showed a moderate to strong positive correlation with BIA (r^2^ ranges between 0.63–0.79). Similarly, in our study, the Slaughter et al. [22] equation also demonstrated a moderately positive correlation with BIA (r = 0.61, *p* < 0.001) and the Wickramasinghe et al. [24] equation (r = 0.59, *p* < 0.001); a low correlation was observed with the Dezenberg et al. [3] equation (r = 0.47, *p* < 0.001). Similarly, in a study conducted in Pakistan among children 9 to 19 years old, the Slaughter skinfold prediction equation accurately predicted %BF at 98.4% accuracy compared to DXA measurement [17], followed by the Dezenberg et al. [25] equation at 90.3%. The observed moderate correlation between BIA and Slaughter equations may in part be explained by age variations in the studies; in our study, besides the inclusion of 8-year-olds, it also included 6- and 7-year-olds, while the Slaughter et al. [22] prediction equations were based on 8-to-29-year-old children and young adults. 

The current study had a homogenous population of Black South African children from the same community. Contrary to the Slaughter et al. [22] and Dezenberg et al. [25] studies, whereby 90% of the children in the Dezenberg et al. [25] equation study were Caucasian, the Wickramasinghe et al. [24] study only focused on an Asian population, while the Slaughter study included white and black children. The Slaughter study does not stipulate the proportion of black vs. white children in the study [22]. In our study, all the prediction equations underpredicted %BF in Black South African children. Even though the age groups used in the three selected prediction equations are within the age ranges of children in the current study, the ethnicity and environment of the study population may have influenced the observed underestimation and prediction in %BF. Previous studies have reported higher fat percentages among Asian children [38,39,40]; thus, the Wickramasinghe et al. [24] equation was developed for Asian children and may influence %BF in a different ethnic group. The differences in %BF were previously reported even within different Asian groups [40]. 

In the current study, girls had higher fat mass, as measured by skinfold and BIA. Similar results were reported in other studies [41,42], with similar age ranges: 4–20 years, 6–9 years, and 7–9 years, respectively. A recent publication from the BC–IT study reported a high prevalence of excessive FM as determined by D_2_O for stable isotope techniques in the same population [33]. The Freedman et al. [18] study also showed higher %BF among girls even though the values were only based on skinfold measurements. Physiological reasoning behind the high body fat in girls is on the basis of fat-deposition aggregates around the hips [8,43] consistent with a wider pelvic structure and breasts, with physical alterations triggered by changing levels of oestrogen, testosterone [43], and growth hormone, along with an increase in the number and size of adipocytes [44].

In our study, the Wickramasinghe et al. [24] prediction showed the most applicable prediction of %BF in the combined group of boys and girls, compared to the BIA equation. Measurements of %BF by both skinfold thickness and BIA have been inconsistent in different studies. A study by Otte and colleagues [45] that compared % BF by BIA and skinfold found a significant difference of 18.50 ± 6.50 and 16.81 ± 6.74, respectively. Contrary, Lubis et al. [46] reflected no difference in %BF between BIA and Skinfold thickness at a population level and between males and females with a *p* = 0.20 for both. 

In the Hastuti et al. [47] study, the skinfold equations by Durnin and Rahaman [48] and that of Wickramasinghe significantly overestimated %BF (*p* < 0.001) in boys and girls, while the equation of Watanabe et al. [49] underestimated the %BF (*p* < 0.001). The equation of Slaughter et al. [22] gave the lowest bias, but it slightly overestimated %BF in boys (−2.02 ± 5.22% difference; *p* < 0.001) and underestimated %BF in girls (0.78 ± 4.48% difference; *p* < 0.001). However, in the Hastuti et al. [47] study, all equations showed a high correlation with %BF obtained from BIA (r = 0.85–0.87; *p* < 0.001) and acceptable range of limits of agreement (LOA; 2.6–5.1%), with the lowest being from Slaughter et al. [22]. These results differ from our study, wherein the Wickramasinge et al. prediction equation [24] had the smallest differences in %BF compared to BIA. Similarly, all predicted equations used in our study underestimated %BF in this Black South African sample. The observed underestimation by Slaughter et al. [22] and Dezenberg et al. [25] skinfold prediction equations was previously recorded by Cameron et al. [50] in a South African study conducted on 9-year-olds aimed at validating the Slaughter et al. [22] and Dezenberg et al. [25] prediction equations using DXA as a reference method. Ethnic and age differences in the populations studied could be the contributory factor to the observed discrepancies. The observed differences may be due to the methods used. The Hastuti et al. [47] study compared the sum of four sites (i.e., triceps, biceps, subscapular, and suprailiac skinfolds) with BIA measurements, while our study used the sum of two skinfold thicknesses (i.e., triceps and subscapular skinfolds), and the age groups as well as the number of participant in the studies differed (12–15 years, n = 610 vs. 6–8 years old, n = 202).

The underestimation of the Slaughter et al. [22] equation has previously been reported in other populations. For example, in a study by Kehoe et al. [41], conducted among 6-to 9-year-old Indian children, the Slaughter equation underestimated %BF only in children with the lowest adiposity. In a study from Chile by Aguirre et al. [42] among children 7 to 9 years old, the Slaughter equation underestimated BF% by −12 and −9% in girls and boys, respectively. In the Aguirre et al. [42] study, the Slaughter et al. [22] equation showed the greatest underestimation of %BF compared to the Ramirez–Zea [51] equations. Similarly, in our study, an underestimation of %BF using the Slaughter equation was −12.65 for girls and −11.92 for boys. Aguirre et al. [51] attributed the underprediction by Slaughter to the fact that the equation was developed in the 80s when obesity was not as prevalent among children as today. The underestimation is observed in children of the same age groups but different ethnic groups.

In the Kehoe et al. [41] study, the Dezenberg et al. [24] equation underpredicted %BF in all the groups, while the Wickramasinghe et al. [24] equations underpredicted in boys and overpredicted in girls. These differences are observed even though our study and their study used the same procedures: the same BIA Bodystat (50 kHz) frequency, the inbuilt manufacturer’s equation, the sum of the subscapular and triceps skinfold thicknesses measurements, and the same age range. The effect of ethnicity (Indian children vs. black children) and environmental factors (India vs. South Africa) may account for the differences observed in the two studies. 

The strength of this study is that it was able to compare %BF derived from secondary methods of skinfold thickness with BIA even though it does not include ‘gold-standard methods’ in a relatively large and homogenous sample of young African children. The non-conclusive results presented in different studies indicate a need for pretesting prediction equations in populations and age groups before adoption. Of importance for this study is that it provided a platform for developing validation studies in this sample. Future studies that explore the validation of these two secondary methods against the criterion method of TBW for the determination of %BF and FFM are needed. 

The limitations of this study are that it did not include different racial groups of South African children. However, the children were selected from one of the major ethnic groups in South Africa, and that may have limited the potential effect of ethnicity. Further, the age range was narrow, only limited to 6–8-year-olds. This study had a homogenous population with similar economic status. Furthermore, the comparison of the two secondary methods and with no reference technique is a limitation of the study. Another limitation of the study is the use of only the sums of two skinfolds measurements. However, in future studies, the inclusion of criterion or ‘gold-standard’ methods will be considered against the two secondary methods and other methods. Furthermore, the inclusion of wide racial and ethnical groups and the broadening of age groups should be considered.

## 5. Conclusions

All three skinfolds equations demonstrated a smaller %BF than the %BF measured by BIA, but the difference is smallest with the Wickramasinghe et al. [24] equation. Based on these findings, it is apparent that the validation and development of equations of %BF against a criterion or ‘gold-standard’ methods for South Africans are urgently needed to refute these findings.

However, in the absence of developed %BF equations or ‘gold-standard’ methods, the available prediction equations are still desirable. 

## Figures and Tables

**Figure 1 ijerph-19-14531-f001:**
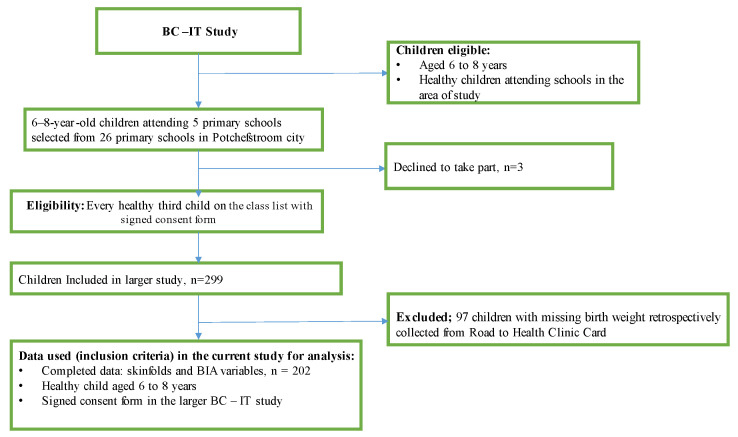
Flow diagram for children included in the study. Caption for Figure 1, demonstrate the procedures followed regarding the participants in the larger BC–IT study which was comprised of 299 participants. Of which in the figure, criteria’s for eligibilities and exclusion to participate in the present study show that out of 299 children 202 met the selection criteria.

**Figure 2 ijerph-19-14531-f002:**
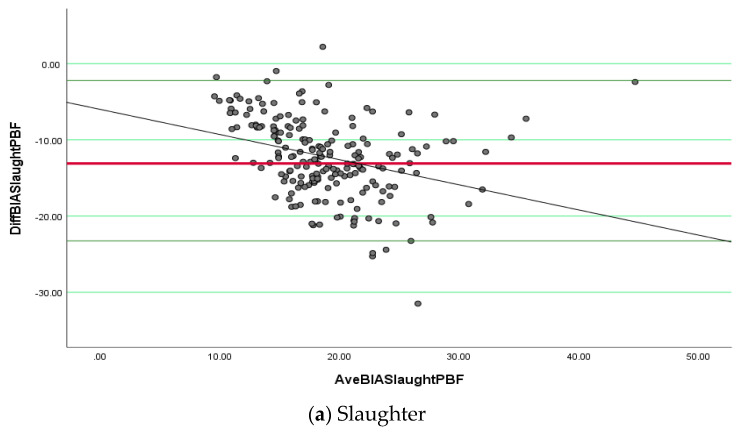
(**a**–**c**): shows the Bland and Altman plots for the mean differences between calculated and measured %BF obtained from BIA and the Slaughter et al. [22], Wickramasignhe et al. [24], and Dezenberg et al. [25] skinfold equations, with upper and lower 95% confidence intervals (CI). [Figure legends: Bland–Altman plots with regression lines of %BF obtained from BIA against those estimated from the skinfold equations of 1(a) Slaughter et al., (b) Wickramasinghe et al., and (c) Dezenberg et al. The y-axis represents the differences between bioelectrical impedance analysis (BIA) and Slaughter prediction equation for body fat (DIFFBIASlaughterBF/Wickramasinghe/Dezenberg), and the x-axis represents the means between BIA and the skinfold prediction equation for body fat DIFFBIASlaughterBF/Wickramasinghe/Dezenberg. The dotted line represents the negative slope, and the solid line represent the mean regression line].

**Table 1 ijerph-19-14531-t001:** Characteristics of skinfold-equation methods used for deriving body fat percentage in children derived from Wickramasinghe, Slaughter, and Dezenberg.

First Author	Reference Method	Reference Method	N	Age (Years)	Ethnicity
Slaughter et al. [22]	%BF (boys) = 1.21 × T(T + SS) − 0.008 × (T + SS)^2^ − 3.2 (African boys) %BF (girls) = 1.33 × (T + SS) − 0.013 (T + SS)^2^ − 2.5	Photo absorptiometry, D_2_O dilution, and hydrostatic weighing	310	8–29	Caucasian and African American
Wickramasinghe et al. [24]	FM (boys) = 0.68 × A + 0.246 × T + 0.383 × SS − 1.61 − 3.45 FM (girls) = 0.680 × A + 0.246 × T + 0.383 × SS − 3.45	D_2_O dilution	188	5–15	South Asia
Dezenberg et al. [25]	FM (boys) = 0.342 × W + 0.256 × T − 6.501 FM (girls) = 0.342 × W + 0.256 × T − 5.501	DXA	202	4–10.9	Caucasian and African American

A = age (months); %BF = percentage body fat; D_2_O = deuterium oxide; T = triceps skinfold (mm); SS = subscapular skinfold (mm); FM = fat mass; DXA = dual-energy X-ray absorptiometry; w = weight (kg).

**Table 2 ijerph-19-14531-t002:** Descriptive characteristics of the total group and for boys and girls.

	Total (n = 202)	Boys (n = 83)	Girls (n = 119)	
	Mean ± SD	Mean ± SD	Mean ± SD	*p*-Value for Sex Differences
Age (year)	7.57 ± 0.85	7.62 ± 0.78	7.54 ± 0.90	0.54
Height (cm)	120.20 ± 7.02	120.83 ± 6.25	119.76 ± 7.50	0.29
Weight (kg)	22.73 ± 4.53	22.69 ± 4.15	22.76 ± 4.80	0.91
BMI (kg/m^2^)	15.61 ± 1.99	15.43 ± 1.73	15.73 ± 2.15	0.29
BMI Z score	−0.17 ± 1.08	−0.29 ± 1.08	−0.09 ± 1.07	0.18
FM (%) BIA	25.12 ± 6.42	22.45 ± 5.51	26.97 ± 6.68	<0.001
FM (kg) BIA	5.79 ± 2.36	5.18 ± 2.02	6.21 ± 2.49	0.002
FFM BIA	16.80 ± 3.07	17.50 ± 2.79	16.31 ± 3.17	0.006
Triceps (mm)	8.20 ± 3.12	7.21 ± 2.38	8.88 ± 3.38	<0.001
Subscapular (mm)	5.93 ± 2.96	5.27 ± 1.75	6.38 ± 3.51	0.01
TBW (BIA) (ℓ)	12.81 ± 2.50	13.34 ± 2.25	12.44 ± 2.61	0.01
%BF using Slaughter equation	12.8 ± 4.89	10.5 ± 3.64	14.2 ± 4.46	0.001
%BF using Wickramasinghe equation	23.2 ± 7.03	17.0 ± 3.88	27.5 ± 5.33	<0.001
%BF using Dezenberg equation	17.8 ± 5.36	17.3 ± 4.69	18.2 ± 5.77	0.252
Age distribution (n (%))				
6 year	55 (27.36)	19 (22.9)	36 (30.3)	0.136
7 year	69 (34.32)	34 (41.0)	35 (29.4)	
8 year	78 (38.80)	30 (36.1)	48 (40.3)	
BMI categories (n (%))				
Underweight (below 18.5 kg/m^2^)	30 (14.9)	19 (22.9)	11 (9.2)	<0.001
Normal weight (18.5 to 25 kg/m^2^)	127 (62.9)	57 (68.7)	70 (58.8)	
Overweight (larger than 25 and less than 30 kg/m^2^)	31 (15.3)	5 (6.0)	26 (21.8)	
Obesity (30 kg/m^2^ or higher)	14 (6.9)	2 (2.4)	12 (10.1)	

%BF = Percentage body fat; BMI = body mass index; FM = fat mass; kg = kilogram; cm = centimeter; mm = millimeters; FFM = fat-free mass; % = percentage; TBW = total body water; n = sample number; BIA = bioelectrical impedance analysis; MD = mean differences, SD = standard deviation.

**Table 3 ijerph-19-14531-t003:** Difference between the percentage body fat (%BF) obtained from bioelectrical impedance analysis (BIA) and the skinfold equations for the total sample.

	Paired Correlations	Paired Differences				
	Mean ± SD	r ± SEE	*p*-Value	MD ± SD	*t*	*p*-Value for the Equation Differences
BIA	25.12 ± 6.42	0.611 ± 5.10	<0.001	−12.35 ± 5.18	−33.896	<0.001
%BF from skinfolds (Slaughter et al. [22])	12.76 ± 4.89					
BIA	25.12 ± 6.42	0.540 ± 5.42	<0.001	−1.92 ± 6.47	−4.217	<0.001
%BF from skinfolds (Wickramasinghe et al. [24])	23.19 ± 7.04					
BIA	25.12 ± 6.42	0.474 ± 5.67	<0.001	−7.27 ± 6.11	−16.910	<0.001
%BF from skinfolds (Dezenberg et al. [25])	17.84 ± 5.36					

MD = mean differences, SD = standard deviation, SEE = standard estimate of error; r = correlation coefficient; t = t-test; *p*-value = determining the difference between BIA and the Slaughter et al. [22], Wickramasinghe et al. [24], and Dezenberg et al. [25] equations.

**Table 4 ijerph-19-14531-t004:** Difference between the %BFs obtained from BIA and various skinfold equations.

	Paired Correlations			Paired Differences		
	Mean ± SD	r ± SEE	*p*-Value	Mean ± SD	*t*	*p*-Value for the Equations
Boys (n = 83)						
%BF BIA	22.45 ± 5.51					
%BF from skinfolds (Slaughter et al. [22])	10.53 ± 3.64	0.484 ± 4.85	<0.0001	−11.92 ± 4.92	−22.08	<0.0001
%BF from skinfolds (Wickramasinghe et al. [24])	17.02 ± 3.88	0.387 ± 5.16	<0.0001	−5.43 ± 5.37	−9.202	<0.0001
%BF from skinfolds (Dezenberg et al. [25])	17.32 ± 4.69	0.397 ± 5.09	<0.0001	−5.13 ± 5.64	−8.279	<0.0001
Girls (n = 119)						
%BF BIA	26.94 ± 6.38					
%BF from skinfolds (Slaughter et al. [22])	14.20 ± 4.46	0.58 ± 5.63	<0.0001	−12.65 ± 5.35	−25.78	<.0001
%BF from skinfolds (Wickramasinghe et al. [24])	27.50 ± 5.33	0.478 ± 5.63	<0.0001	0.52 ± 6.05	0.947	0.345
%BF from skinfolds (Dezenberg et al. [25])	18.20 ± 5.77	0.516 ± 5.20	<0.0001	8.77 ± 6.00	−15.937	<0.0001

%BF = Percentage body fat; BIA = bioimpedance analysis; r = correlation coefficient; SEE = standard estimate of error; t = t-test; SD = standard deviation.

**Table 5 ijerph-19-14531-t005:** Limits of agreement between the percentage body fat (%BF) obtained from bioelectrical impedance analysis (BIA) and various prediction equations.

	Boys (n = 83)		Girls (119)	
	Mean ± Limit	Lower; Upper	Mean ± SD	Lower; Upper
BIA vs. %BF from skinfolds (Slaughter et al. [22])	−11.92 ± 4.92	−12.99; −10.85	−12.65 ± 5.35	−13.62; −11.68
BIA vs. %BF from skinfolds (Wickramasinghe et al. [24])	−5.43 ± 5.37	−6.60; −4.25	0.52 ± 6.05	−0.57; 1.62
BIA vs. %BF from skinfolds (Dezenberg et al. [25])	−5.13 ± 4.64	−6.36; −3.89	−8.77 ± 6.00	−9.86; −7.68

%BF = percentage body fat; n = sample number; SD = standard deviation.

## Data Availability

The datasets used for analyses during the current study are not publicly available due ethical restrictions and participant confidentiality but are available from the corresponding author on reasonable request and in accordance with the NWU data sharing policy.

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
