# Peer review of "Comparison of Several Prediction Equations Using Skinfold Thickness for Estimating Percentage Body Fat vs. Body Fat Percentage Determined by BIA in 6–8-Year-Old South African Children: The BC–IT Study"

_ijerph, 2022, doi:10.3390/ijerph192114531_

Round 1
Reviewer 1 Report
General Comment:
Overall, this is a well written paper regarding to a research topic which there is sparse information in African population in such ages. The manuscript is well written, and the introduction is clearly providing the necessary rationale for the research question. Well done for the research process! Below I present minor suggestions.
Methods:
Has normality been verified before any other test? Which test has been applied for normality? This information should be clear on the statistical descriptors before any test.
Discussion
I also suggest to authors to highlight the details about the importance of this results, and it practice relevance/practical application of the study and its results.
Author Response
Author's Reply to the Review Report (Reviewer 1)
General Comment:
Overall, this is a well written paper regarding to a research topic which there is sparse information in African population in such ages. The manuscript is well written, and the introduction is clearly providing the necessary rationale for the research question. Well done for the research process! Below I present minor suggestions.
Response: Thank you for your positive comments.
Methods:
Has normality been verified before any other test? Which test has been applied for normality? This information should be clear on the statistical descriptors before any test.
Response: We provided clarification in terms of how normality was verified as, the normality of the data was done using the Normality distribution of the data was determined with the Kolmogorov–Smirnov test and graphical methods, i.e. histograms and Q-Q plots. We performed independent t-test for normally distributed data to determine sex differences and chi-squared test for age and BMI categories
Discussion
I also suggest to authors to highlight the details about the importance of this results, and it practice relevance/practical application of the study and its results.
Response: In our revised manuscript we added these statements ‘The findings in this study highlighted the need to be careful when using already available equation when calculating %BF in children. As such, the combination of the skinfolds predictions for %BF and BIA may be reduce the likelihood of misdiagnosis of high or low lean mass in children.

Reviewer 2 Report
This study compares BIA with different equations that estimate body fat percentage from manual anthropometry. This paper is well-written and easy to read. However, BIA might not be an appropriate reference method for body composition. The accuracy of BIA could be also changed by applying equations to participants of different ages or races. Using BIA as a reference to compare other methods cannot determine the accuracy. The author should consider using laboratory methods such as DXA to evaluate the equations of manual anthropometry or clarify the accuracy of BIA equations used on their target group.
Abstract
The reference style is not consistent in the abstract and main context.
Introduction
Line 91: The equations were selected because of the skinfold sites. This sounds like the authors collected the data and then determine the testing equations. This needs further clarification. There might be some equations that used further skinfold sites and provide accurate predictions.
Line 94: The author indicated that BIA provided unreliable data but used it as the reference method in this study. This needs further clarification.
Some illustration is required to indicate how the target participants in this study differ from the existing literature.
Materials and Methods
Line 168 This sentence seems not to belong to the current section.
Results:
Table3 Further illustrations are required to clarify to distinguish the difference in p-values.
Discussion:
Further detail should be provided to clarify the effects (Line 389)
The potential reason for the comparison result should be discussed and investigated further (e.g. Line, 398; Line403). The current context just shows the difference between studies but does not deliver the meaning of this difference. Is there any trend of this difference? How should future research know from these differences?
Others:
Some information shows the template contexts (e.g. Supplementary Materials) should be edited.
Author Response
Author's Reply to the Review Report (Reviewer 2)
Comments and Suggestions for Authors
This study compares BIA with different equations that estimate body fat percentage from manual anthropometry. This paper is well-written and easy to read. However, BIA might not be an appropriate reference method for body composition. The accuracy of BIA could be also changed by applying equations to participants of different ages or races. Using BIA as a reference to compare other methods cannot determine the accuracy. The author should consider using laboratory methods such as DXA to evaluate the equations of manual anthropometry or clarify the accuracy of BIA equations used on their target group.
Response: We agree with the review that BIA has limitations, and that is the reasons in lines 415 & 416 acknowledging the limitations of BIA we indicated that ‘Future studies that explore the validation of these two secondary methods against criterion method of TBW for the determination of %BF and FFM are needed.’ The reviewer recommendation regarding to the use of laboratory is receiving attention in which in our future study validation will be done.
Comment:
Abstract
The reference style is not consistent in the abstract and main context.
Response: Corrections has been made; and thank you.
Comments:
Introduction
Line 91: The equations were selected because of the skinfold sites. This sounds like the authors collected the data and then determine the testing equations. This needs further clarification. There might be some equations that used further skinfold sites and provide accurate predictions.
Response: In our revised manuscript clarification has been made as ‘Guided by published literature, the skinfold prediction equations by Slaughter et al. [1] Wickramasighe [2] and Dezenberg [3], were selected because of the age range and the skinfold sites used in the validation studies the methods used.’
Comment:
Line 94: The author indicated that BIA provided unreliable data but used it as the reference method in this study. This needs further clarification.
Response: In our revised manuscript a clarification has been made as ‘However, González-Ruíz et al. [19] stated that the use of BIA (an easy-to-use and low-cost method for the estimation of fat-free mass (FFM) in healthy individuals) in low-resources communities as an alternative should be used with caution given the fact that BIA prediction models differ according to the sample's characteristics in which they have been derived. Even though BIA is not a criterion method its validation against existing skinfolds predictions in South African children is hardly studied.’
Comment:
Some illustration is required to indicate how the target participants in this study differ from the existing literature.
Response: The participants in the current manuscript are not different from the larger BC – IT study because they are homogenous.
Comments:
Materials and Methods
Line 168 This sentence seems not to belong to the current section.
Response: Clarification has been made in the revised manuscript as ‘The characteristics of skinfold equations methods used for deriving %BF in children de-rived from Wickramasinghe, Slaughter and Dezenberg are presented in Table 1.’, hence amendments are effected on the table title and now it reads as ‘Table 1. Characteristics of skinfold equations methods used for deriving body fat percentage in children derived from Wickramasinghe, Slaughter and Dezenberg’.
Results:
Comments:
Table3 Further illustrations are required to clarify to distinguish the difference in p-values.
Response: In our revised manuscript we clarified the p-value as follows p-value for several equations difference; and went on to provide a footnote in the table as ‘p-value = determining the difference between BIA and Slaughter, Wickramasinghe, and Dezenberg equations’.
Discussion:
Further detail should be provided to clarify the effects (Line 389)
Response: Clarification has been made in the sentence, and the sentence read as ‘The effect of ethnicity (Indian children vs black children) and environmental factors (India vs South Africa) may account for the differences observed in the two studies.’
Comments:
The potential reason for the comparison result should be discussed and investigated further (e.g. Line, 398; Line403). The current context just shows the difference between studies but does not deliver the meaning of this difference. Is there any trend of this difference? How should future research know from these differences?
Response:
In our revised manuscript clarification has been made as ‘The observed differences may be due to the methods used. The Hastuti et al. [43] study compared the sum of four sites (i.e. triceps, biceps, subscapular and suprailiac skinfolds) with BIA measurements, while our study used the sum of two skinfold thickness (i.e. triceps and subscapular skinfolds), and the age groups as well as the number of participant in the studies differ (12–15 years, n=610 vs 6–8 years old, n=202).
Others:
Some information shows the template contexts (e.g. Supplementary Materials) should be edited.
Response: Since we do not have supplementary materials in the submission of the manuscript the sub-section Supplementary Material has been deleted in the main manuscript.
Round 2
Reviewer 2 Report
Thanks for the authors revising the paper. However, most of my comments cannot be addressed. A further revision to provide a strong rationale and clear illustration is necessary.
The authors agree that BIA is not an appropriate validation method. Therefore, it is required to provide a strong rationale for why it is worth comparing BIA and skinfolds. Validation should not be used in the context as BIA is not a reference method. Many repetitions in the context, especially in the Discussion section. Editing is required.
Introduction
The author did not answer why to use these two sites for skinfolds instead of others. The equation selection criteria need to address clearly.
The author should illustrate why it is required to compare BIA and skinfolds for South African children. What is the advantage to understand the comparison of BIA and skinfolds? What is the specific of South African children compared to the participants in other groups in previous studies?
Discussion
I suggest making a table to compare the result between current and previous studies.
The potential effect of ethnicity and environmental factors should be addressed further. How do these factors affect body composition? What should future researchers expect if they are testing South African black children?
The author should provide practical suggestions to illustrate how future research should be noticed if there are similar differences happen.
Author Response
Reviewer #2 Round 2
Comments and Suggestions for Authors
Thanks for the authors revising the paper. However, most of my comments cannot be addressed. A further revision to provide a strong rationale and clear illustration is necessary.
Response: Thank you. In our revised manuscript we addressed provided more arguments in strengthening the rationale for this study.
Comments:
The authors agree that BIA is not an appropriate validation method. Therefore, it is required to provide a strong rationale for why it is worth comparing BIA and skinfolds. Validation should not be used in the context as BIA is not a reference method. Many repetitions in the context, especially in the Discussion section. Editing is required.
Response: As with the response above, motivation for the study has been improved. In terms of the raised comment regarding repetitions in the discussion, corrections have been made.
Introduction
Comments:
The author did not answer why to use these two sites for skinfolds instead of others. The equation selection criteria need to address clearly.
Response: In our revised manuscript clarification regarding the use of two sides of skinfolds and prediction equations are made.
Comments:
The author should illustrate why it is required to compare BIA and skinfolds for South African children. What is the advantage to understand the comparison of BIA and skinfolds? What is the specific of South African children compared to the participants in other groups in previous studies?
Response: We provided the reasons for comparison of BIA and skinfolds in our South African children by providing illustrations of what is currently available in the literature. Of which in the conclusion of the study we concluded that ‘All three equations demonstrated a smaller %BF than the %BF measured by BIA, but the difference is smallest with the Wickramasinghe equation. Based on these findings, it is apparent that validated and development of equations of %BF against a criterion or ‘gold standard’ methods for South African are urgently needed to refute these findings.
However, in the absence of developed %BF equations or ‘gold standard’ methods the available prediction equations are still desirable.’
Discussion
Comments:
I suggest making a table to compare the result between current and previous studies.
Response: We think the table 4 in the results section is relevant and it address this comment. Besides this table, in our discussion a through comparative studies in line with our findings has been made.
Comments:
The potential effect of ethnicity and environmental factors should be addressed further. How do these factors affect body composition? What should future researchers expect if they are testing South African black children?
Response: Amendments have been made in the revised manuscript.
Comments:
The author should provide practical suggestions to illustrate how future research should be noticed if there are similar differences happen.
Response: In our revised manuscript we provided suggestion which followed our conclusion as ‘All three equations demonstrated a smaller %BF than the %BF measured by BIA, but the difference is smallest with the Wickramasinghe equation. Based on these findings, it is apparent that validated and development of equations of %BF against a criterion or ‘gold standard’ methods for South African are urgently needed to refute these findings.
However, in the absence of developed %BF equations or ‘gold standard’ methods the available prediction equations are still desirable.’
